# The Harmful Effects of Welding Fumes on Human Dental Enamel—A Microhardness Analysis

**DOI:** 10.3390/dj12100332

**Published:** 2024-10-17

**Authors:** Catrinel Ștefania Petrovici, Răzvan Alexandru Grăjdeanu, Adina Petcu, Monica Vasile, Beatrice Marcela Severin, Doru Florin Petrovici, Lucian Cristian Petcu

**Affiliations:** 1Doctoral School of Medicine, Ovidius University, 900573 Constanţa, Romania; catrinel.petrovici@gmail.com; 2Dental Technician, CoraCERAM Dental Laboratory, 900394 Constanța, Romania; rzv.g@icloud.com; 3Department of Physics, Faculty of Pharmacy, Ovidius University, 900470 Constanţa, Romania; 4Department of Radiotherapy, Ovidius Clinical Hospital, 905900 Constanţa, Romania; lucian-cristian.petcu@365.univ-ovidius.ro; 5Department of Physics, Faculty of Medicine, Ovidius University, 900470 Constanţa, Romania; 6Department of Health Management, Faculty of Medicine, Ovidius University, 900470 Constanţa, Romania; beatrice.severin@univ-ovidius.ro; 7Department of Community Oral Health, Faculty of Dentistry, Ovidius University, 900684 Constanţa, Romania; doru.petrovici@365.univ-ovidius.ro; 8Department of Biostatistics, Faculty of Medicine, Doctoral School of Medicine, Ovidius University, 900470 Constanţa, Romania

**Keywords:** dental enamel, microhardness, welding fumes, Vickers method, indentation

## Abstract

**Introduction:** Over the years, welding fumes’ harmful effects have been demonstrated countless times in the scientific literature. Recently, studies in the field have shown an increasing interest in the negative consequences that these fumes may have on the tissues of the oral cavity. **Materials & method:** The current study aimed to investigate the impact that welding fumes have on the structure of human dental enamel by analyzing the microhardness of the dental enamel in 15 extracted human teeth, after various exposure times, using the Vickers method. **Results:** The results obtained after 48, 96, 168, and 336 h of direct exposure of the extracted specimens to the welding fumes show a statistically significant increase in the depreciation of the dental enamel’s microhardness, related to the duration of exposure (*p* < 0.05). An average of 305 Vickers units was observed at the longest exposure time, 336 h, in the present study, whereas in the control group, the microhardness analysis showed an average of 327 Vickers units.

## 1. Introduction

The toxicity of welding fumes (WFs) has been repeatedly demonstrated in the scientific literature, and the harmful effects they have on the health of workers in the field have been studied in detail. The effects of harmful substances resulting from welding processes range from acute respiratory/neurological symptoms (“metallic smoke fever,” migraines, etc.), to chronic pathologies, which are developed by prolonged exposure to the fumes (examples include bronchitis and lung cancer) [1].

Welding processes generate pollutants that pose significant risks to the environment and also pose a danger in the workplace. Metal particles and fumes resulting from welding operations are risk factors for workers’ health: both long-term and short-term exposures to welding vapor have been linked to a variety of adverse effects on general health in epidemiological studies.

Recent scientific discoveries, made during epidemiological investigations, cross-sectional studies, and case report studies, focus on the correlation between the metals used in different welding processes, health conditions, and emerging diseases, serving as a statement to advance preventive measures that influence and benefit the health of welders [2,3].

Among a variety of health risks affecting workers in this field, there is one that stands out from the others, namely, the impact of fumes resulting from the welding process on the respiratory system, affecting the upper respiratory tract [1].

Contaminants of the environment (of any kind—chemical, biological, or physical) have this ability to distort, in order to modify the genome of the human organism, causing mutations at this level. As previously demonstrated in the scientific literature, occupational exposure to these contaminants occurs in different ways (direct contact with the skin, ingestion, or inhalation—the last being the most common in the case of people who sweat) [4]. The toxicity mechanisms of chromium (hexavalent—has the highest toxic potential) and nickel can be either direct or indirect and can be described by enzymatic inhibition and the determination of oxidative stress at a cellular level. The substratum of these mechanisms consists of the generation of free radicals, through the interaction of metabolites in the water in the body with the metal particles that have reached the body, thus favoring damage to the structure of DNA by breaking the chains and causing chromosomal aberrations with their intersection, thereby causing cell death [4].

The mechanisms of cellular damage through contact with metals in welding processes have not yet been clarified, but the body’s reactivity to them has been demonstrated repeatedly [4]. What is known is that hydroxyl radicals could interfere in the defense and repair mechanisms of DNA and RNA molecules, proteins and lipids, and the membranes of nuclei and mitochondria [5,6]. These events could be considered responsible for initiating carcinogenesis, teratogenesis, and premature cell aging. As far as carcinogenesis is concerned, for example, cytotoxicity can lead to the development of chronic tissue trauma; cellular hyperplasia has emerged as a compensatory phenomenon of cellular multiplication, which can subsequently lead to tumor transformation [6].

It follows that there is a connection between the cytotoxicity of WFs and damage to the human genome, and the accumulation of toxic factors can contribute to the malignant transformation of normal cells [4]. The biological markers used in the present study have also been used for the investigation of the risk of cancer of the digestive tract and the upper respiratory tract, for which there is still a possible link between the genotoxicity of weld smoke and its contact with the oral mucosa [4].

However, a concrete correlation between the occurrence of oral mucosal cancer and exposure to WFs has not yet been scientifically demonstrated; this study comes as a further confirmation of previous scientific studies demonstrating the toxicity of chronic exposure to welding fumes compared to findings for those who are not in contact with these contaminants [4,5]. 

Since the oral cavity is structured not only by soft tissues but also by dental elements, which are considered hard structures, more scientific studies on the phenomena that may occur as a result of contact between dental enamel and WFs are required. This should be done in order to describe a complex and complete picture of the diseases caused by such fumes on the whole organism in general, thus taking into account the teeth as components of the oral system.

The experimental in vitro study described here aimed to associate these WFs with the presence of damage to the hardness of the dental structures, resulting from their inhalation, following prolonged exposure through the work environment; this situation has been briefly described in the literature [7], although the scientific data are not conclusive and are too limited in this regard. Currently, there are studies in the literature describing the link between WFs and the soft tissues of the oral cavity [8] and, the urge to discover even more harmful effects of these welding fumes is growing in the dentistry field.

This study targets to examine the influence of welding fumes on the structure of human dental enamel by analyzing the microhardness of dental enamel in 15 extracted teeth after exposure durations of 48, 96, 168, and 336 h using a specially designed testing device for WF submission. The microhardness analysis of the enamel samples was performed utilizing the Vickers indentation method. The teeth included in the study were selected only from male patients, considering the fact that the welding industry is mostly dominated by male workers, due to the vicious work environmental conditions.

## 2. Materials and Methods

This study was carried out on 15 teeth that had recently been extracted (from male patients that underwent dental treatment in the PerfectDent1 Dental Clinic in Constanța, Romania, and were diagnosed with advanced chronic marginal parodontopathy or with orthodontic conditions which required extraction of certain teeth in order to obtain the necessary space for the start of the orthodontic treatment), imposing the exclusion criterion for the experiment, whereby any dental structures affected by caries were not taken into account for testing, so as not to distort the results obtained. Figure 1 shows a succinct sample analysis of the teeth included in the study, based on the age of the patients.

After extraction, the teeth were cleaned with distilled water and were then stored in containers with saline solution, at an ambient temperature of 20–22 °C, until the time of preparation. They were then tested in vitro to observe the effect of WFs on the hardness of the enamel.

A testing device was designed in CoraCeram Dental Laboratory, Constanta, Romania, to simulate, as accurately as possible, the conditions of contact of the teeth with the fumes resulting from welding processes (Figure 2). The testing device and method have a few limitations, as not all the normal homeostatic conditions from inside the oral cavity could be reproduced, such as oral humidity due to saliva presence.

### 2.1. Description of the Testing Device

The testing device is composed of the following components:Testing chamber—a container with both input and output in order to fill it up with WFs and adjust the air pressure inside (Appendix A Figure A1);A relative pressure manometer (pressure gauge) to keep track of the air pressure inside the testing chamber (Appendix A Figure A2);A vacuum pump, which is used to reach negative pressure inside the testing chamber, enabling the capture of WFs inside the testing container (Appendix A Figure A3);A vacuum manometer to monitor the negative pressure inside the testing chamber (Appendix A Figure A4);Two faucets to adjust the pressure;An electronic valve to maintain a constant pressure inside the testing chamber;An electronic fan to recirculate the contaminated air flux inside the testing chamber;A heating resistance to increase the temperature inside the testing chamber to simulate the homeostatic temperature from inside the oral cavity (approximately 37 °C);A temperature probe for long-term measurements of the temperature inside the testing unit;A thermostat connected to the temperature probe and to the heating resistance to set up the testing temperature (approximately 37 °C) (Appendix A Figure A5);A carbon dioxide tank with a pressure reducer, which is used to create the mixture of WFs (Appendix A Figure A6);A device for capturing the WFs and directing them into the testing container;A fixing support for the tooth probes to be tested;A support plate to put together the entire assembly.A TIG welding machine (Parkside, PTMI 180) TIG/MMA

For detailed pictures of the components, see Appendix A (Figure A1, Figure A2, Figure A3, Figure A4, Figure A5 and Figure A6).

### 2.2. Description of the Testing Procedure

The operating principles of this testing device consist of the following steps, which are based on reproducing the working environmental conditions of a welding worker:

Step 1. The tooth to be tested was fixed in the dedicated support.

Step 2. The vacuum pump was connected to create negative pressure inside the testing chamber to −0.6 bar.

Step 3. The welding arc was initiated using a rutilic electrode and a galvanized steel bar, followed by the production of the WFs. The inlet valve to the testing chamber was open throughout this procedure.

Step 4. The WFs were collected using the dedicated device and were directed inside the testing container; there was up to a 0 value of pressure inside the chamber.

Step 5. The carbon dioxide tank was connected and gas was released inside the testing chamber, using the pressure reducer valve up to the value of 0.5 of the relative pressure manometer.

Step 6. Waiting time for each test.

During this procedure, the introduction of “fresh” WFs every 24 h was required. The study considered an average of 4 h of exposure to WFs/working day of 8 h. The following figures show the aspect of the extracted teeth after they had been exposed to welding fumes for several periods of time (48 h, 96 h, 168 h, and 336 h) using the methodology and testing device presented in the previous section (Figure 3, Figure 4, Figure 5 and Figure 6). 

### 2.3. Preparing Tooth Enamel Samples for Microhardness Testing

The 15 male patients’ teeth (both exposed and unexposed to WFs) were sectioned lengthwise and crosswise (Appendix A Figure A7) using a diamond disc motor to obtain clear cuts of the cusps or the incisal edge (Appendix A Figure A8). The enamel cuts were then fixed into acrylic resin (Appendix A Figure A9) and were polished using ultrafine granulation discs, in order to obtain clean-cut edges and very smooth surfaces of the specimens (Appendix A Figure A10 and Figure A11); after each polishing course, the specimen probes were thoroughly washed with distilled water.

For detailed explanations about the samples’ preparation, see Appendix A (Figure A7, Figure A8, Figure A9, Figure A10 and Figure A11).

From all the specimens obtained and prepared for testing, 25 probes were randomly selected; they were divided into five groups, as follows:Group A = {G1-A, G2-A, G3-A, G4-A, M-A}
Group B = {G1-B, G2-B, G3-B, G4-B, M-B}
Group C = {G1-C, G2-C, G3-C, G4-C, M-C}
Group D = {G1-D, G2-D, G3-D, G4-D, M-D}
Group E = {G1-E, G2-E, G3-E, G4-E, M-E},
where

G1-A, -B, -C, -D, and -E were specimens exposed for 48 h to WFs, simulating 10 working days for a welder;

G2-A, -B, -C, -D, and -E were specimens exposed for 96 h to WFs, simulating 20 working days for a welder;

G3-A, -B, -C, -D, and -E were specimens exposed for 168 h to WFs, simulating 40 working days for a welder;

G4-A, -B, -C, -D, and -E were specimens exposed for 336 h to WFs, simulating 80 working days for a welder;

M-A, -B, -C, -D, and -E were specimens unexposed to WFs.

### 2.4. Analysis of Enamel Microhardness for Teeth Exposed to WFs

The microhardness analysis of the teeth exposed to WFs was conducted using the Vickers method. Each research specimen had to be pressed with a diamond indenter at a particular load and for a particular amount of time, as part of the process. After the force was removed, the diagonals (d 1 and d 2) of the indentation were measured. Each specimen’s hardness was then determined using the following equation (Figure 7):

The indentations made in each sample were made using a CV-400 AAT Micro-Vickers hardness tester (CV Instruments, Brasov, Romania) (Figure 8), as shown in the following representation (Figure 9), and the imprints were interpreted using the dedicated software Test Engineer for HV.

Each enamel sample was indented 15 times, resulting in 15 hardness imprint measurements for each sample. Figure 10 shows the indentation process for enamel specimens from each test group.

### 2.5. Statistical Analysis for Enamel Microhardness Values After WF Exposure

The results obtained from microhardness analysis were statistically interpreted using one-way ANOVA, the Shapiro–Wilk test, and Student–Newman–Keuls post hoc analysis for multiple comparisons. 

For a better understanding, the results obtained after analyzing the enamel’s microhardness after exposure to the WFs are systematically included in Table 1, Table 2 and Table 3 and Figure 11 and Figure 12.

For each group, the results of the one-way ANOVA show that there are statistically significant differences between at least two of the mean hardness values of the compared samples (Table 2).

## 3. Results for Enamel Microhardness Analysis After WF Exposure

According to the post hoc analysis (Student–Newman–Keuls test) applied to each group, it can be stated that there are statistically significant differences in mean hardness values between all samples of each group (*p* < 0.05). The longer the duration of exposure, the more the value of the tooth enamel hardness is depreciated compared to the test hardness value.

All samples from control group M and all samples from groups G1, G2, G3, and G4 were compared with each other. For each group, the results of the one-way ANOVA show that there are no statistically significant differences (*p* > α = 0.05) between the mean hardness values of the compared samples (Table 3).

Considering the results presented in Table 3, the data were grouped into five groups, M, G1, G2, G3, and G4; each group had 75 values. The results of the one-way ANOVA show that there are statistically significant differences between at least two of the mean hardness values of the compared samples (F = 500.571, *p* < 0.001 < α = 0.05). According to the post hoc analysis (Student–Newman–Keuls test), it can be stated that there are statistically significant differences regarding the mean hardness values between all samples (*p* < 0.05). The longer the exposure time, the more the tooth enamel hardness value decreases compared to the hardness value of the control sample M (Figure 11).

## 4. Discussion

Exposure to industrial fumes has long been debated in the scientific world, a discussion that began as a result of global industrialization for the development of technology. In this regard, the most recent study conducted in the field of exposure to fumes resulting from welding processes dates from the year 2022; it was in the form of a systematic review and retrospective cohort study over a period of 25 years in patients who developed squamous cell carcinoma of the oropharyngeal and oral mucosa, and it originated from the United Kingdom [9]. It is known that the cancers of the head and neck are often correlated with the factors smoking (as a social habit) and infection with H-papilloma virus, but in the absence of this connection, scientists have decided to study in detail the causes of the occurrence of oral and pharyngeal carcinomas for patients with these pathologies. 

In 2020, Barul et al. demonstrated a clear connection between exposure to weld smoke and the risk of developing squamous cell carcinomas with oropharyngeal localization or at the end of the brain [10]. With regard to the exposure of teeth to the toxicity of these WFs, studies are limited, and it is precisely this that has led to the need to closely explore these effects on tough dental structures, in order to draw the attention of dental specialists to the potential for dental pathologies at the level of dental enamel.

Unlike the usual exposure to tobacco smoke that affects the hardness of the enamel to a more limited extent, whereby the appearance of signs of dental damage takes longer to produce changes for the welding smoke, a result of 305 Vickers units was obtained after an exposure of 336 h, simulating about 80 days of work for a welder (which can be considered relatively fast for a career that can start from the age of 18 years) [11]. 

Another study that highlights the toxicity of atmospheric fumes in the working field was conducted in 2017, which tracked the concentration of heavy metals in the structure of the teeth, which was taken as a biomarker for the risk of fume exposure. The authors found relatively high concentrations of As, Mn, Ba, Cu, Cr, Pb, Zn, and other elements that are also found in the welding industry, without having a symmetrical distribution; some of these were found in increased concentrations at the level of the molars, while others were found at the level of the incisors. The basic idea in this study, which is similar to other studies in the literature, although providing a summary in terms of damage to teeth by industrial toxins, is that the longer the time of exposure (taking as a criterion the age of the subjects included in the study), the higher the concentrations of heavy metals. One of the conclusions of this study was that the tough structure of the tooth can be considered a true biological indicator of industrial pollution [12]. 

The toxicity of fumes of any nature, industrial or customary, has been studied closely with the structure of the teeth and in the study carried out by Ibrahim and Hassan. In 2020, their work once again demonstrated that these toxins have clear negative effects on the ultrastructure of the enamel, affecting even the natural process of remineralization but also the mechanical qualities of the hard substance (microhardness) [13]. Other results obtained in relation to exposure times are consistent with the studies in the literature; for the specimens exposed for 168 h, the value of microhardness in Vickers units was 311, while for those exposed for 96 h, a value of 314 was obtained, and for the samples exposed to WFs for 48 h, a value of 320 was obtained. This is without statistically significant differences from the control group included in the study, for which values of 327 were measured (as mean values for each lot exposed). 

It should be noted, once again, that the entire experiment was designed and executed so as to simulate as closely as possible the real situation of inhalation of fumes produced as a result of the welding process, with the rendering of pressure and temperature conditions in the oral cavity and the external environment. 

The limitations that could be noted for this study are related to the thickness of the enamel layer of the teeth included in the study: thickness varied from one tooth to another and even from one dental surface to another for the same tooth. Another limitation of this study may be that the study was carried out in vitro, and it is known that teeth outside the oral cavity may have different reactivities to various stimuli than those tested in vivo.

## 5. Conclusions

The results obtained through this study are concordant with the average results declared in the current scientific literature regarding the toxicity of fumes (especially tobacco exhaled fumes), and the situation is very similar to the contact between enamel and tobacco smoke and the contact between enamel and WFs. 

After a 48-h exposure to welding smoke, there was no statistically significant difference in dental enamel hardness compared to that for the control group. However, after 336 h of exposure (equivalent to about 80 working days), the enamel hardness was noticeably lower, at an average of 305 Vickers units, compared to 327 units in the control group. The intermediate stages of these exposure intervals were 96 h and 168 h, equivalent to approximately 20 and 40 working days for a welding worker, respectively.

This test was conducted to raise awareness about the toxicity of WFs, which have been shown in multiple scientific studies to have harmful effects on respiratory health. Dentists may encounter unique erosive lesions that do not seem to be related to known factors but could be linked to a patient’s dental condition and the environment in which they work. 

Regulating exposure to toxins from welding fumes falls within the jurisdiction of labor protection and competent authorities in the occupational health department. 

They are responsible for ensuring the use of high-quality equipment that minimizes the inhalation of toxins. Implementing measures to minimize exposure to WFs can improve the overall health and quality of life for industrial workers. 

## Figures and Tables

**Figure 1 dentistry-12-00332-f001:**
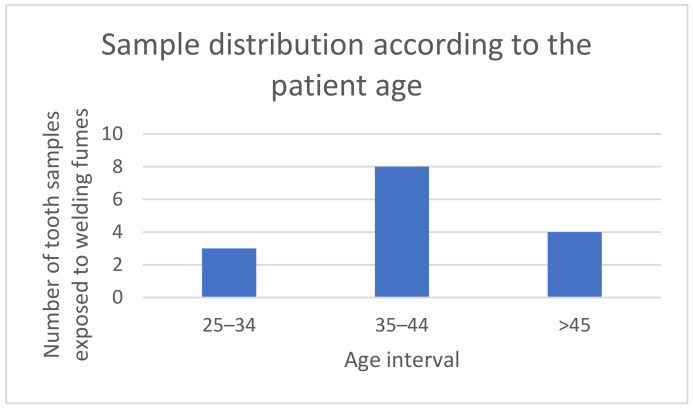
Distribution of tooth samples subjected to welding fumes, according to the age of the patients.

**Figure 2 dentistry-12-00332-f002:**
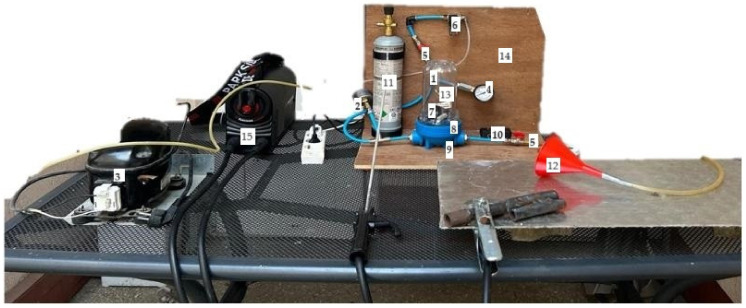
Test device for contact between teeth and WFs.

**Figure 3 dentistry-12-00332-f003:**
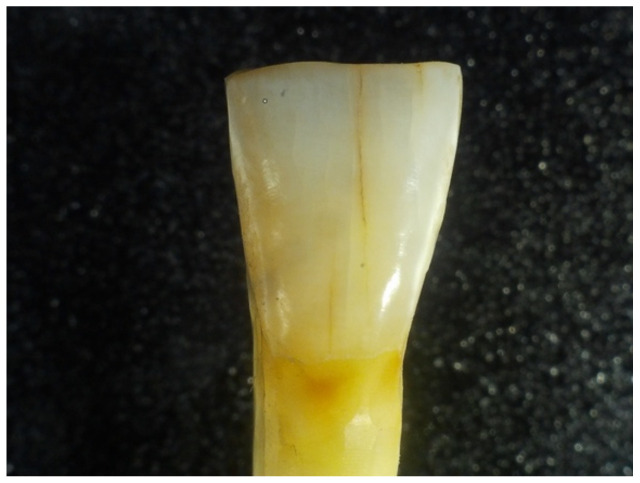
Aspect of tooth sample after a 48-h exposure to WF.

**Figure 4 dentistry-12-00332-f004:**
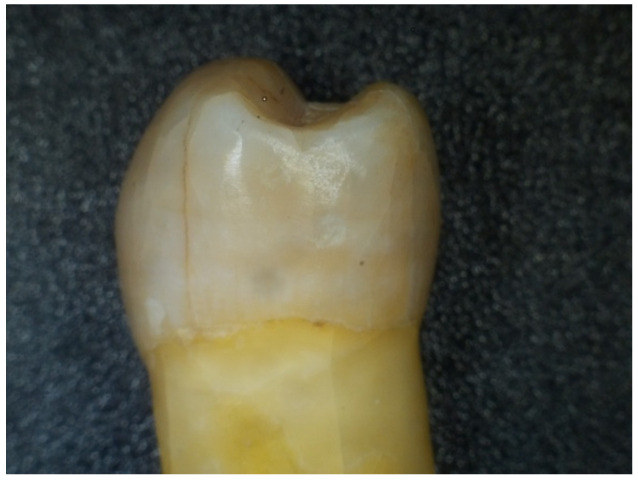
Aspect of tooth sample after a 96-h exposure to WF.

**Figure 5 dentistry-12-00332-f005:**
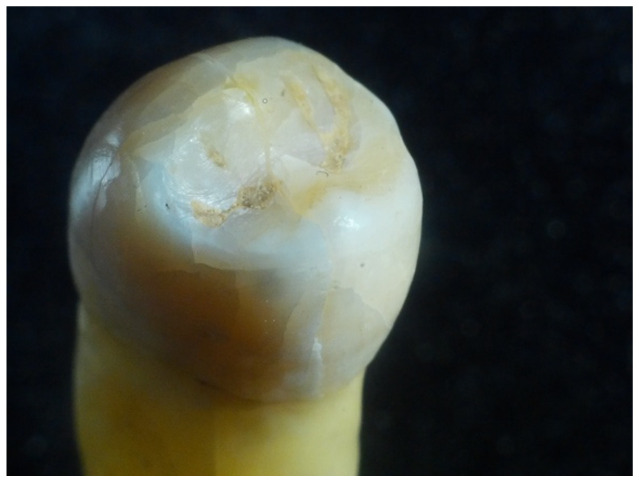
Aspect of tooth sample after a 168-h exposure to WF.

**Figure 6 dentistry-12-00332-f006:**
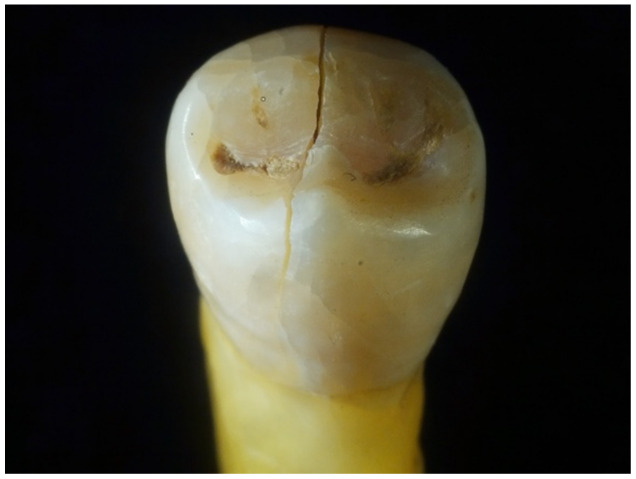
Aspect of tooth sample after a 336-h exposure to WF.

**Figure 7 dentistry-12-00332-f007:**
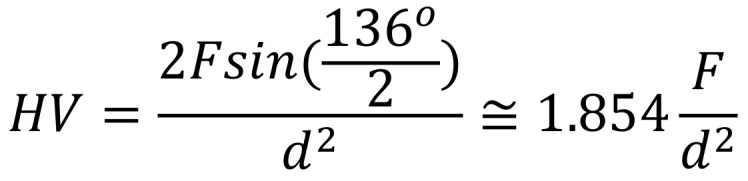
Equation for enamel microhardness measurements, where F = load in Kgf, and d = average between d 1 and d 2.

**Figure 8 dentistry-12-00332-f008:**
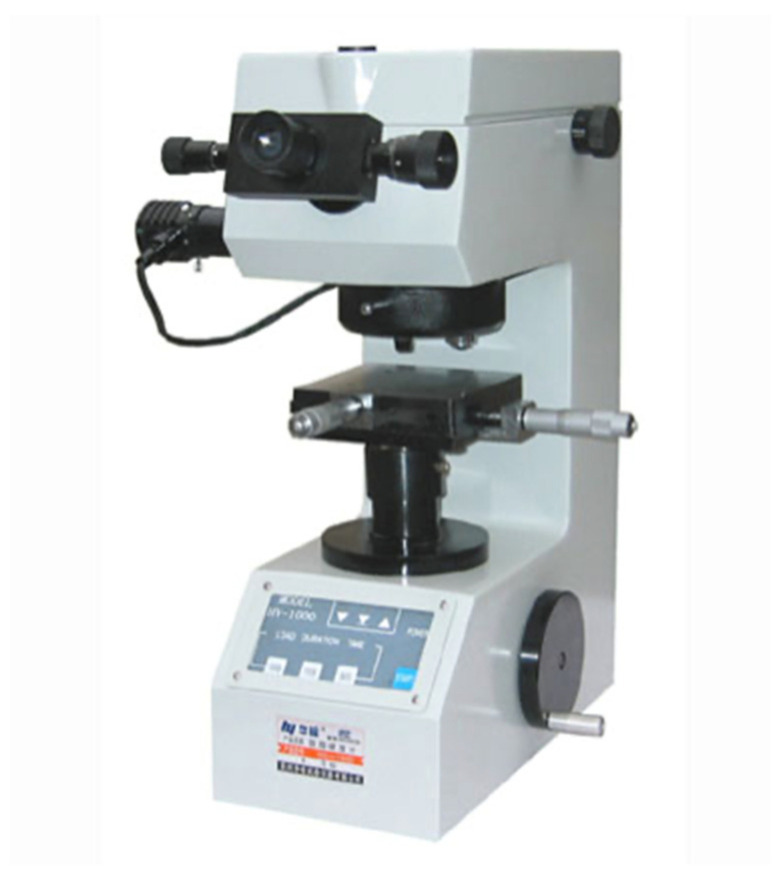
Micro-Vickers hardness tester CV-400 AAT.

**Figure 9 dentistry-12-00332-f009:**
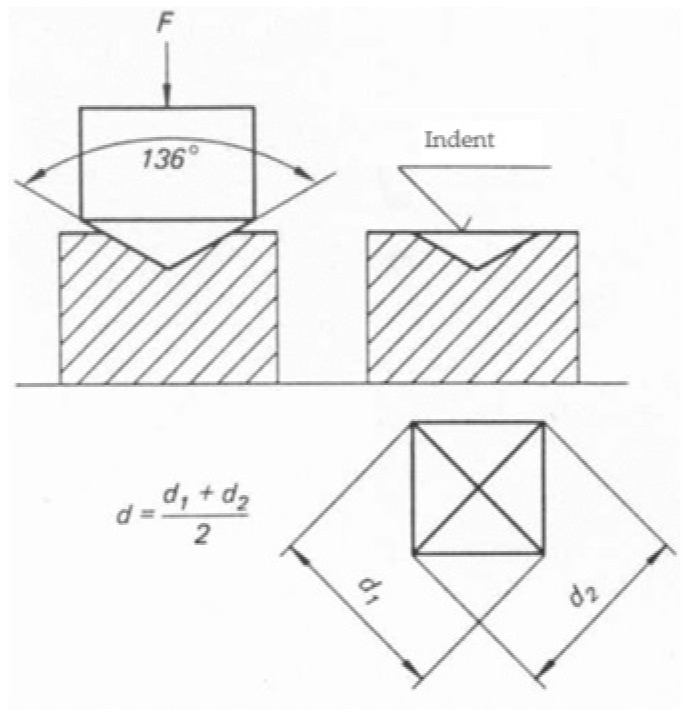
Schematic representation of imprint made on dental enamel specimens.

**Figure 10 dentistry-12-00332-f010:**
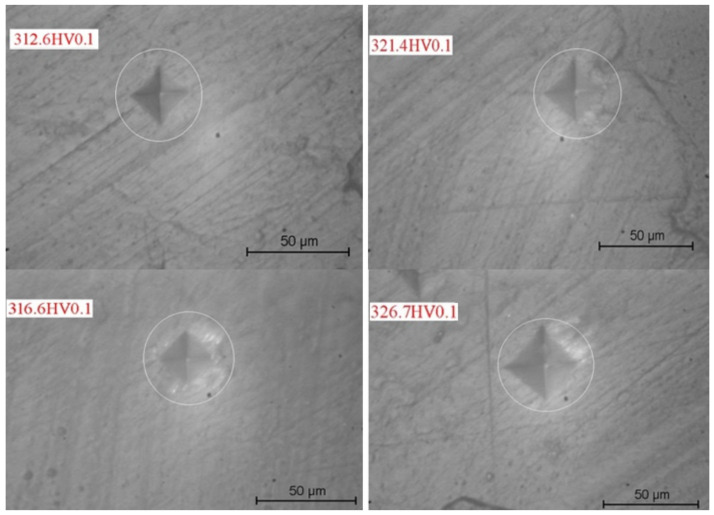
Indentation measurements on dental enamel specimens exposed to WF.

**Figure 11 dentistry-12-00332-f011:**
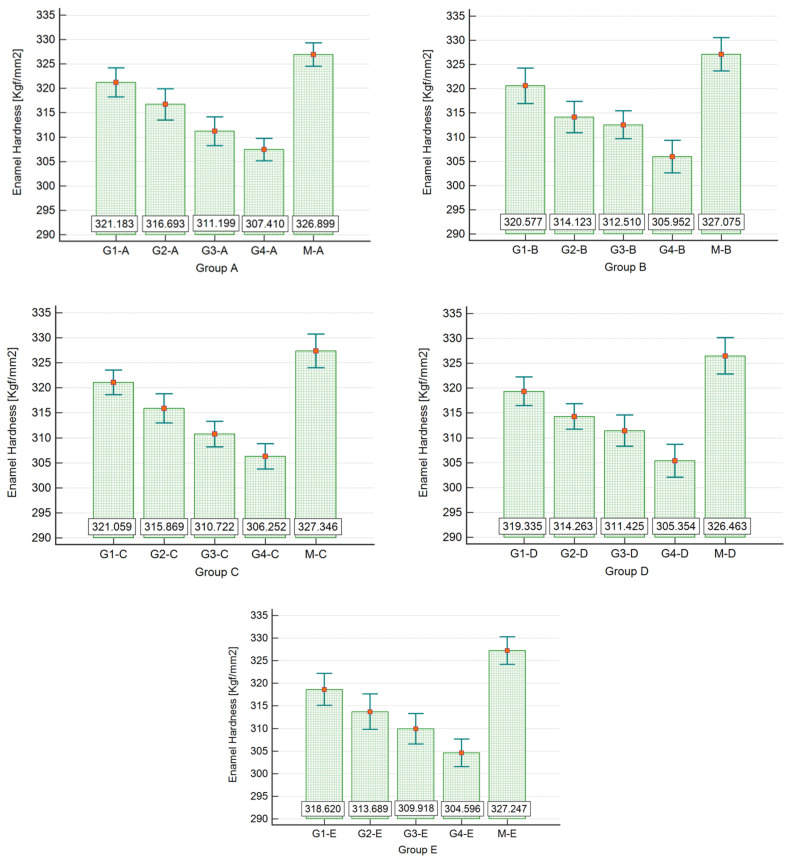
Bar and error bar chart (mean ± SD) for dental enamel hardness values in Groups A to E; time = 10 s, load = 100 gf.

**Figure 12 dentistry-12-00332-f012:**
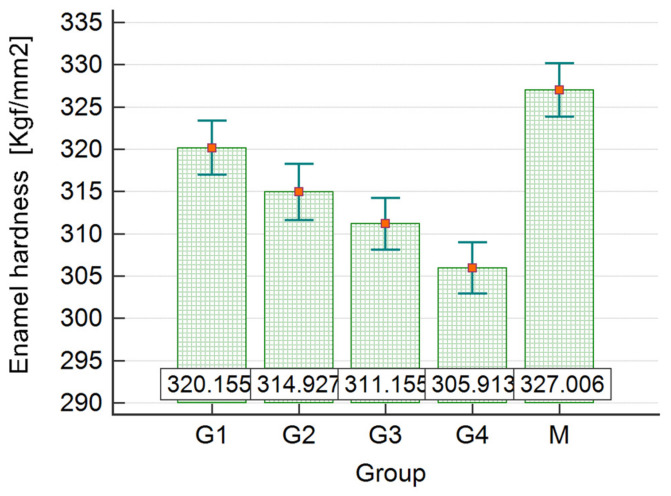
Bar and error bar chart (mean ± SD) for dental enamel hardness values in Groups M, G1, G2, G3, and G4; time = 10 s, load = 100 gf.

**Table 1 dentistry-12-00332-t001:** Summary statistics—dental enamel hardness [Kgf/mm^2^], Groups A to E.

	Dental Enamel Hardness [Kgf/mm^2^], Group A				
	G1-A	G2-A	G3-A	G4-A	M-A
N	15	15	15	15	15
Mean	321.183	316.693	311.199	307.410	326.899
SD	2.958	3.192	2.948	2.332	2.420
Minimum	315.230	311.600	307.110	303.781	322.894
Maximum	325.000	321.200	316.110	312.230	331.489
Shapiro–Wilk test	W = 0.9077	W = 0.9359	W = 0.9181	W = 0.9748	W = 0.9769
	*p* = 0.1250	*p* = 0.3338	*p* = 0.1800	*p* = 0.9211	*p* = 0.9442
	Dental Enamel Hardness [Kgf/mm^2^]—Group B				
	G1-B	G2-B	G3-B	G4-B	M-B
N	15	15	15	15	15
Mean	320.577	314.123	312.510	305.952	327.075
SD	3.674	3.214	2.876	3.386	3.454
Minimum	314.250	309.600	308.240	301.220	320.340
Maximum	326.370	318.920	317.560	311.434	332.692
Shapiro–Wilk test	W = 0.9728	W = 0.9183	W = 0.9596	W = 0.9413	W = 0.9684
	*p* = 0.8967	*p* = 0.1816	*p* = 0.6861	*p* = 0.3996	*p* = 0.8336
	Dental Enamel Hardness [Kgf/mm^2^]—Group C				
	G1-C	G2-C	G3-C	G4-C	M-C
N	15	15	15	15	15
Mean	321.059	315.869	310.722	306.252	327.346
SD	2.447	2.918	2.572	2.531	3.361
Minimum	317.250	310.250	306.580	302.554	321.436
Maximum	325.370	319.920	315.664	310.110	332.876
Shapiro–Wilk test	W = 0.9627	W = 0.9620	W = 0.9633	W = 0.9322	W = 0.9587
	*p* = 0.7391	*p* = 0.7264	*p* = 0.7490	*p* = 0.2945	*p* = 0.6704
	Dental Enamel Hardness [Kgf/mm^2^]—Group D				
	G1-D	G2-D	G3-D	G4-D	M-D
N	15	15	15	15	15
Mean	319.335	314.263	311.425	305.354	326.463
SD	2.869	2.567	3.143	3.323	3.678
Minimum	314.250	310.600	307.240	302.110	319.340
Maximum	323.370	318.780	317.110	311.110	331.898
Shapiro–Wilk test	W = 0.9477	W = 0.9429	W = 0.9410	W = 0.8289	W = 0.9632
	*p* = 0.4891	*p* = 0.4207	*p* = 0.3958	*p* = 0.0089	*p* = 0.7470
	Dental Enamel Hardness [Kgf/mm^2^]—Group E				
	G1-E	G2-E	G3-E	G4-E	M-E
N	15	15	15	15	15
Mean	318.620	313.689	309.918	304.596	327.247
SD	3.524	3.930	3.381	3.037	3.048
Minimum	314.100	308.450	305.120	299.120	321.665
Maximum	326.622	320.920	316.980	309.908	331.658
Shapiro–Wilk test	W = 0.8954	W = 0.9443	W = 0.9624	W = 0.9695	W = 0.9506
	*p* = 0.0810	*p* = 0.4395	*p* = 0.7332	*p* = 0.8506	*p* = 0.5344

**Table 2 dentistry-12-00332-t002:** Output of ANOVA for Groups A to E.

	Test Statistics	Probability
Group A = {G1-A, G2-A, G3-A, G4-A, M-A}	F = 115.947	*p* < 0.001
Group B = {G1-B, G2-B, G3-B, G4-B, M-B}	F = 87.942	*p* < 0.001
Group C = {G1-C, G2-C, G3-C, G4-C, M-C}	F = 133.754	*p* < 0.001
Group D = {G1-D, G2-D, G3-D, G4-D, M-D}	F = 97.383	*p* < 0.001
Group E = {G1-E, G2-E, G3-E, G4-E, M-E}	F = 96.896	*p* < 0.001

**Table 3 dentistry-12-00332-t003:** Output of ANOVA for Group M and Groups G1, G2, G3, and G4.

	Test Statistics	Probability
Group M = {M-A, M-B, M-C, M-D, M-E}	F = 0.175	*p* = 0.950
Group G1 = {G1-A, G1-B, G1-C, G1-D, G1-E}	F = 1.948	*p* = 0.112
Group G2 = {G2-A, G2-B, G2-C, G2-D, G2-E}	F = 2.433	*p* = 0.059
Group G3 = {G3-A, G3-B, G3-C, G3-D, G3-E}	F = 1.516	*p* = 0.207
Group G4 = {G4-A, G4-B, G4-C, G4-D, G4-E}	F = 1.896	*p* = 0.121

## Data Availability

Data are available upon request to the corresponding authors.

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
