# Peer review of "The Harmful Effects of Welding Fumes on Human Dental Enamel—A Microhardness Analysis"

_dentistry, 2024, doi:10.3390/dj12100332_

Round 1
Reviewer 1 Report
Comments and Suggestions for Authors
The manuscript does not clearly explain why WF is relevant to enamel damage. It is essential to provide background information on the role of WF in enamel degradation to justify the study. I suggest that you include a section in the introduction that discusses the connection between WF and enamel damage, supported by relevant literature. Explain how WF potentially affects enamel and why it is significant to investigate this relationship. The manuscript lacks references to previous research related to WF and enamel damage. Situating your study within the context of existing literature is crucial.
The aim of the study is not clearly stated. Clear objectives are necessary to guide the reader and clarify the purpose of your research. Please explicitly state the aim of your study in the introduction section.
In the methodology section, restructuring the device components and procedural steps into clear, straightforward sentences to enhance understanding and facilitates ease of following the procedure.
The material and methods section needs to be rewritten, as it appears to have been written randomly and haphazardly. It is advisable to explain testing procedure and enamel preparation under two separate heading. Additionally, including study strengths and limitations, generalizability of the study, and practical values will enhance the manuscript. Unnecessary contents are included in multiple areas of the manuscript such as this statement ‘’This study was conducted in association with the Biophysics Department of the Fac- 163 ulty of Medicine, Ovidius University of Constanta and in collabooration with 164CORACERAM – DentalTech Laboratory in Constanta, Romania.
The writing style needs improvement, and several texts require rephrasing. The study lacks clear hypotheses and theoretical backgrounds. Therefore, it is recommended to provide operational definitions of key terms and abbreviations.
Comments on the Quality of English LanguageThe writing style needs improvement, and several texts require rephrasing.
Author Response
Comment 1:
The manuscript does not clearly explain why WF is relevant to enamel damage. It is essential to provide background information on the role of WF in enamel degradation to justify the study. I suggest that you include a section in the introduction that discusses the connection between WF and enamel damage, supported by relevant literature. Explain how WF potentially affects enamel and why it is significant to investigate this relationship. The manuscript lacks references to previous research related to WF and enamel damage. Situating your study within the context of existing literature is crucial.
Response 1: Thank you for your dedicated time towards reviewing our manuscript. We agree with your statement. The primary idea of studying the connection between welding fumes and their potential toxicity on human enamel came from treating several patients (welding workers) along the years of practice, in Constanta, Romania, patients that presented abnormal affects of the enamel surface "erosion-like", but that did look differently from any other type of erosion described in the dentistry literature. Therefore, we have decided to take the matter a step forward into the direction of studying if there is truly a connection between WFs and enamel damage. As the scientific literature is very brief in this regard, we had not terms of comparison, most of the existing studies describing only a link between the toxicity of WFs and the presence of oral carcinoma, or other abnormal cell development. We also have tried our best to find and we did add other scientific studies that could support our testing theory.
Comment 2: "The aim of the study is not clearly stated. Clear objectives are necessary to guide the reader and clarify the purpose of your research. Please explicitly state the aim of your study in the introduction section."
Response 2. Oral health should be regarded as a whole and hard tissues should be inspected thoroughly and the aim of this experimental in vitro study is to pinpoint this connection, as the welders' community is a very important part of our city industry, as a city port at the Black Sea. We have made some modifications, accordingly, in the Introduction regarding the aim of the study.
Comment 3:
"In the methodology section, restructuring the device components and procedural steps into clear, straightforward sentences to enhance understanding and facilitates ease of following the procedure.
The material and methods section needs to be rewritten, as it appears to have been written randomly and haphazardly. It is advisable to explain testing procedure and enamel preparation under two separate heading. Additionally, including study strengths and limitations, generalizability of the study, and practical values will enhance the manuscript. Unnecessary contents are included in multiple areas of the manuscript such as this statement ‘’This study was conducted in association with the Biophysics Department of the Fac- 163 ulty of Medicine, Ovidius University of Constanta and in collabooration with 164CORACERAM – DentalTech Laboratory in Constanta, Romania."
Response 3: Thank you for pointing this out! We have accordingly restructured the MATERIALS & Method section, by dividing it into multiple subsections and also updated the figures number to make it more comprehensive. Therefore, we would like to invite you to also take a look in the Appendix section A for more detailed images of the project methodology (pages 12 to 16 of the manuscript).
Comment 4: "The writing style needs improvement, and several texts require rephrasing. The study lacks clear hypotheses and theoretical backgrounds. Therefore, it is recommended to provide operational definitions of key terms and abbreviations."
Response 4: Your firm suggestion was taken into consideration and we took action by applying for the MDPI Academic and Language Editing Services. The current form of the manuscript should now be more scientific appealing and easy-to-read for a fellow reader for that matter.
Thank you for your prompt response and kind support!
Sincerely,
Dr. Catrinel Petrovici
Reviewer 2 Report
Comments and Suggestions for Authors
Congratulations, the study is very interesting
Author Response
Comment 1: "Congratulations, the study is very interesting"
Response 1: We would like to express our sincere gratitude for your prompt response! It is indeed a very interesting study, considering the fact that the theme of our project has been briefly discussed in the present scientific literature, and by approaching that matter we truly hope to open new topics of interest towards the subject.
Sincerely,
Dr. Catrinel Petrovici
Reviewer 3 Report
Comments and Suggestions for Authors
The method is described, however, the article does not address the control of other variables present in the oral environment, such as gender, age, saliva, diet, lifestyle, etc., which are relevant in the daily life of an exposed worker. These variables are crucial to consider in order to determine the actual harmful effects on teeth.
Cancerous squamous cells are mentioned in the discussion. Throughout the article, the method does not focus on such conditions, so it is not clear why they are referenced in the discussion.
It is important to justify the control of the variables mentioned.
Author Response
Comment 1: "The method is described, however, the article does not address the control of other variables present in the oral environment, such as gender, age, saliva, diet, lifestyle, etc., which are relevant in the daily life of an exposed worker. These variables are crucial to consider in order to determine the actual harmful effects on teeth."
Response 1: Thank you for your observation. In the primary manuscript we missed out on mentioning that the 15 sample teeth were extracted from male patients and we rephrased the explanation in the current revised form of the paper. As for other variables, those could not be reproduced entirely, as our project is an experimental in vitro study and we only took into consideration for testing the teeth samples that were completely undamaged by other external factors (such as cavities, demineralisation lesions, traumatic lesions, etc.). We are aware of the limitations our testing method might have, still, this experimental project only aimed to see if there is truly a connection between the toxicity of WFs and the microstructure of the human dental enamel.
Comment 2: "Cancerous squamous cells are mentioned in the discussion. Throughout the article, the method does not focus on such conditions, so it is not clear why they are referenced in the discussion."
Response 2: We understand your point and we would like to clarify that the discussions on the oral carcinoma where presented in the section only to highlight the toxicity of the welding fumes and their effects that may occur after being chronically exposed to those noxious substances, in the process of inhalation.
Comment 3: "It is important to justify the control of the variables mentioned."
Response 3: As we stated before and also mentioned in the revised form of the manuscript, not all the variables could be controlled during our experimental in vitro method. We only accepted full healthy teeth and tried to replicate the homeostatic temperature from inside the oral cavity, for it is well known that extreme variations of the temperature could damage the enamel surface.
Thank you again for all your observations and suggestions, hoping that our response would give you some clarity towards our study protocol.
Sincerely,
Dr. Catrinel Petrovici